# Biological Effect of Soy Isoflavones in the Prevention of Civilization Diseases

**DOI:** 10.3390/nu11071660

**Published:** 2019-07-20

**Authors:** Marzena Pabich, Małgorzata Materska

**Affiliations:** Department of Chemistry, Faculty of Food Science and Biotechnology, University of Life Sciences in Lublin, Akademicka 15 Street, 20-950 Lublin, Poland

**Keywords:** isoflavones, soy, civilization diseases

## Abstract

Scientific advancements in recent years have shed new light on the relationship between diet and human health. Nutrients play an important role in the prevention of many civilization diseases, such as osteoporosis, type II diabetes, hypercholesterolemia, and cardiovascular diseases. The biological activity of natural plant components allows their use in the treatment of various diseases, especially civilization diseases, to be speculated. Special attention is paid to phenolic compounds that have numerous health-promoting properties. Isoflavones, phenolic compounds, are commonly found in legumes, especially in soybeans. Their structural similarity to 17-β-estradiol (E2), the main female sex hormone, allows them to induce estrogenic and antiestrogenic effects by binding to estrogen receptors, and their consumption has been associated with a decreased risk of hormone-related cancers. In addition, numerous epidemiological studies and related meta-analyses suggest that soy consumption may be associated with a lower incidence of certain diseases. However, there are some doubts about the potential effects on health, such as the effectiveness of cardiovascular risk reduction or breast cancer-promoting properties. The purpose of this review is to present the current knowledge on the potential effects of soy isoflavone consumption with regard to civilization diseases.

## 1. Introduction

Significant advancements in science and technology have not only developed civilization, but have also led to the occurrence of many unknown threats that cause diseases. These threats include civilization diseases mainly caused by polluted environments, adulterated and unhygienic food, poor nutrition, and stress due to a fast-paced life. In highly developed and developing countries, health conditions that have a great social impact include obesity, diabetes, cardiovascular diseases, hypertension, osteoporosis, and cancer [1]. Most of these diseases could be potentially counteracted by providing proper nutrition. A balanced diet contains abundant natural substances and ingredients that exert a beneficial effect on the functioning of the human body [2]. For example, polyphenolic compounds are secondary metabolites produced by plants and they show positive effects in the treatment and prevention of civilization diseases. They also provide protection against ultraviolet radiation, fungi, and pathogenic bacteria. Depending on the number of aromatic rings and the way in which they are combined, they are divided into the following classes: flavonoids, phenolic acids, stilbene, and lignans. The most abundant polyphenols are flavonoids, and their division into subclasses depends on the degree of oxidation of the pyran ring: flavonols, flavones, isoflavones, flavanones, anthocyanins, flavanols, catechins, chalcones, aurones, and others [3,4]. Polyphenols have been proved to be important for both plant and human health. Humans and animals cannot synthesize polyphenols and, therefore, humans consume polyphenols along with plant foods as part of their diet [5]. Polyphenols are found in fruits (including nuts), vegetables, legumes, cereals, and flowers, as well as in beverages (e.g., coffee and tea) and chocolates [4].

Soy and the isoflavones it contains are gaining more and more attention because of the health benefits associated with their consumption. The reviews discussed in the following texts address the relationship between isoflavones and civilization diseases. Numerous studies have suggested that soy isoflavones provide protection against hormone-dependent cancers (including breast and prostate cancer (PCa)), osteoporosis, cardiovascular diseases, and diabetic conditions, and alleviate the symptoms associated with the menopause [6,7]. A study showed that the compounds present in soy had a positive effect on human health and thus suggested an increase in the amount of dietary supplements containing isoflavones and food fortification with these compounds [8].

## 2. Metabolism of Isoflavones

Isoflavones mainly occur in plants in the form of biologically inactive glycosides (e.g., genistin and daidzein). After ingestion, they are hydrolyzed in the gut by intestinal bacterial β-glucosidases, resulting in the formation of appropriate bioactive aglycones (e.g., genistein and daidzein) [9,10]. Daidzein can be metabolized to dihydrodaidzein and then to equol or O-desmethylangolensin (O-DMA) (Figure 1). In turn, genistein can be hydrolyzed to dihydrogenistein and then to 6ʹ-hydroxy-O-DMA (6ʹ-OH-O-DMA), which in turn can be degraded to p-ethylphenol [11]. In addition, the fermentation process increases the content of the more bioavailable aglycones. Daidzein, genistein, equol, and O-DMA are the major isoflavones detected in the blood and urine of animals and humans [12]. Nevertheless, only a fraction of aglycones can directly enter the circulation and become available to all other cells of the body. They persist in the plasma for ~24 h, with an average half-life of 6–8 h [13]. Daidzein demonstrates a higher bioavailability than genistein because it has a longer half-life in the intestine. Less-absorbed genistein degrades about twice as fast [14].

Due to the structural similarity of isoflavones to 17-β-estradiol (Figure 2), they are also called phytoestrogenes. Isoflavones may bind to both estrogen ERα and ERβ receptors, thus exhibiting both estrogenic and antiestrogenic properties. However, isoflavones have a low estrogenic potency compared to estradiol [16]. The distribution of ERα and ERβ receptors depends on the tissue types. Reproductive tissues, especially those of the uterus and breast, are abundant in estrogen receptors. 17-β-estradiol shows the same level of affinity toward ERα and ERβ receptors, whereas isoflavones show a greater affinity toward ERβ receptors. Binding of the phytoestrogen to the receptor may result in partial activation of the receptor (agonistic effect) or displacement of the estrogen molecule, thereby reducing receptor activation (antagonistic or antiestrogenic effect) [17]. Researchers are interested in tissue selective phytoestrogens because antiestrogenic activity in reproductive tissues can help reduce the risk of hormone-related tumors (e.g., breast, uterus, and prostate), while estrogenic activity in other tissues can help maintain the bone mineral density and improve blood lipid profiles [18].

## 3. Consumption of Soy

The consumption of soy and soy products varies among countries of the world (Figure 3). In Asian countries, where soy is the most popular vegetable, the recommended range is 20–50 g of soy per day [7,19]. In the United States and most Western countries, its consumption is less popular.

The consumption of soy is related to the daily intake of isoflavones—the highest intake of isoflavones is in the East and South Asian countries (20–50 mg/day). In the United States, it is 0.15–3 mg and in the European countries it is 0.49–1 mg [21,22]. Differences in the intake of isoflavones are undoubtedly related to the fact that in Western countries, a lot of meat, carbohydrates, and highly processed foods, and far less leguminous products, including soy, are consumed. In Poland, soy and the products obtained from it are also not very popular and are consumed in small amounts. 

In addition, epidemiological studies have shown that the incidence of femoral neck fractures in older Asian women was lower than that in Caucasian women [23]. The incidence of cancers was also lower among the population of eastern Asia than that of Western countries [24]. This is associated with a higher consumption of soybeans and the major role it plays in the diet of the inhabitants of Asian countries.

## 4. Isoflavones in Type II Diabetes Mellitus (DM)

Epidemiological studies have indicated that the increased intake of dietary soy isoflavones positively correlates with a lower incidence of diabetes and increased tissue sensitivity to insulin [25]. Although numerous studies (Table 1) have proved the antidiabetic action of isoflavones or soy extracts, the mechanism underlying these beneficial effects is still largely unknown. 

Ademiluyi et al. [26] found that the phenolic compounds present in soy have the ability to inhibit the activity of α-amylase and α-glucosidase (enzymes closely involved in the hydrolysis of carbohydrates in the gastrointestinal tract) based on in vitro studies. As a result, they can slow the absorption of glucose in the small intestine and prevent postprandial hyperglycemia. However, they pointed out that phenolic compounds from the soy extract can regulate glucose absorption by means of mechanisms other than inhibiting the activity of those enzymes. 

Based on the results of studies conducted on obese rats with type II diabetes, the authors suggested that soy isoflavones exert a beneficial effect on lipid and glucose metabolism by activating peroxisome proliferator-activated receptors (PPAR). These receptors regulate the transcription of genes involved in lipid and glucose homeostasis and lipid metabolism in the cell [27]. PPARγ plays a crucial role in glucose metabolism and insulin sensitization, which are usually the molecular targets for certain antidiabetic drugs [28,29]. 

In a double-blind, placebo-controlled cross-over study conducted by Jayagopol et al. [25], the effect of soy protein supplementation (30 g/day) containing 132 mg of phytoestrogens on insulin resistance, glycemic control, and cardiovascular markers in postmenopausal women with type 2 diabetes (T2D) was determined. These studies have shown that short-term phytoestrogen supplementation reduces insulin resistance and improves glycemic control in postmenopausal women with T2D, while reducing the risk of cardiovascular events by lowering low-density lipoprotein cholesterol. The authors also stated that it is necessary to conduct longer studies to determine whether these effects are permanent and have a beneficial effect on the reduction of cardiovascular events.

A prospective, population-based study of 64,227 middle-aged Chinese women found that higher intakes of legumes, in particular soy, were associated with a reduced risk of type 2 DM. In addition, the authors did not find any interaction between the menopause status, total consumption of soy protein or soy, and risk of T2D [34]. 

Supplementation with isoflavones in the concentrations consumed by the Asian population (160 mg) of healthy postmenopausal women living in the Baltimore metropolis did not affect the markers of glucose homeostasis in spite of the increase in the level of adiponectin in the serum. The authors suggested that the lack of beneficial effects on metabolic parameters may have resulted from the fact that the women participating in the study were healthy and their body weight was close to normal. They also emphasized that the antidiabetic potential of isoflavones in obese postmenopausal women requires further research [35]. 

Soy consumption is generally low in Western populations, which limits the ability of epidemiological studies to determine the relationship between soybean intake and T2D. Therefore, most clinical trials concern Asian residents (Table 1). 

Ding et al. [28] examined the association between the urinary excretion of isoflavonoids and risk of T2D. They conducted a nested case–control study among 1111 T2D patients who provided urine samples and were free of diabetes, cardiovascular disease, and cancer during urine sample collection. For this purpose, they measured the urinary excretion of daidzein and genistein, as well as their metabolites O-DMA, dihydrogenistein, and dihydrodaidzein. The authors observed inverse associations between the urinary excretion of daidzein and genistein and risk of T2D in the American women. In addition, the inverse association for daidzein was stronger in postmenopausal women who did not use hormone replacement therapy [28]. Talei et al. [36] did not find a significant association between urine phytoestrogen metabolites and the risk of T2D in middle-aged and elderly Chinese residing in Singapore.

## 5. Isoflavones and Osteoporosis

In recent years, many studies have been conducted on the importance of phytoestrogens in the prevention and treatment of osteoporosis, which is currently a major health problem around the world. WHO defines it as a systemic skeletal disease characterized by a low mass and reduced quality of bone tissue, which results in increased bone fragility. In addition, it belongs to the group of 21st-century diseases. Often, it is called a “silent thief” because it does not cause chronic pain and the first symptom is mostly a fracture. Osteoporosis mainly affects postmenopausal women and older men. It is estimated that it occurs in 30% of women and 8% of men over 50 years of age [37]. In women with osteoporosis, the main reason is the menopause, which causes a sudden reduction in estrogen levels. Deficiency of this hormone results in an increased bone turnover rate and leads to an imbalance in bone remodeling; that is, the resorption rate is slightly greater than the formation of bone tissue, thus accelerating the loss of bone mass [38]. To slow down the process of bone loss in women with estrogen deficiency, hormone replacement therapy (HRT) is used. Due to the numerous side effects and the risk of thrombophlebitis, breast cancer, stroke, or coronary heart disease, HRT should not be recommended as the first-line therapy in the prevention and treatment of osteoporosis. Therefore, soy isoflavones have found a use as an alternative source of estrogens in peri-and postmenopausal women [23,39,40,41]. 

The basis of the protective action of isoflavones in the process of maintaining bone density in postmenopausal women may be related to the affinity of phytoestrogens toward β receptors, which determines their ability to inhibit osteoclast resorption activity and stimulate bone osteoblastic activity [17]. Measurement of bone mineral density loss (BMD) using dual-energy X-ray absorptiometry is considered a “gold standard” among imaging methods of BMD and is usually used in the diagnosis of osteoporosis [42]. 

In an in vivo study carried out by Tit et al. [41], the effectiveness of soy isoflavones and HRT in the prevention of postmenopausal osteoporosis in women after physiological menopause was compared. For this purpose, the BMD and urinary concentration of deoxypyridinoline were determined. After 12 months, it was observed that both therapies had a beneficial effect on bone metabolism, thereby resulting in a significant reduction in the bone resorption process. These results are consistent with previous literature reports. Morabito et al. [43] found that after 12 months of genistein administration, the increase in bone density was comparable to the effect of HRT with estrogen. 

A study carried out by Turhan et al. [44] has shown that six-month administration of isoflavones (59.6 mg genistein with 15.6 mg daidzein) to postmenopausal women led to a significant increase in BMD in the lumbar spine and a concurrent reduction in the biochemical markers of bone resorption. Similar conclusions were obtained by Lee et al. [45], who examined the effects of isoflavone supplementation on bone biomarkers in 87 postmenopausal women. The results suggested that supplementation of 70 mg of isoflavones daily for 12 weeks had a positive effect on bone formation markers: the bone alkaline phosphatase (BALP) alkaline fraction increased by 6.3 ± 4.1% and osteocalcin by 9.3 ± 6.2%. In addition, these results are in line with the meta-analysis of randomized controlled trials, which showed that isoflavones significantly increase serum BALP levels (20.3%) when used at a dose of 75 mg/day [40]. 

In a previous study, women at an early menopause stage were supplemented with 66 mg of isoflavones and soy protein, and the results showed a decrease in the level of bone resorption markers. In addition, cardiovascular indices were also found to be improved in this group [46]. Based on these results, the authors suggested that the addition of isoflavones to the Western diet may significantly reduce the number of women diagnosed with osteoporosis. The number of osteoporosis cases in Western European countries is high, which is related to the low consumption of soy and its products, compared to Asian countries, where soy is the main component of the diet. It was also found that the Western diet, which is rich in animal protein, promotes the acidification of body cells and increases the excretion of urinary calcium. As a result, the availability of calcium for metabolic processes is reduced, which contributes to the development of osteomalacia and osteoporosis [47,48].

## 6. Hormone-Associated Cancers

Cancer diseases are the biggest challenge of modern medicine. Despite the significant progress that has been made in recent years, the incidence and mortality rates of cancer are still increasing worldwide. Breast cancer is currently one of the largest epidemiological problems in women. Based on GLOBOCAN estimates, about 2.1 million newly diagnosed female breast cancer cases and over 600,000 breast cancer deaths occurred in 2018 [49]. Therefore, in recent times, phytoestrogens have attracted attention in the context of chemoprevention due to their very broad spectrum of biological activity. 

Isoflavones exhibit weak estrogenic activity due to their steroid structure. Genistein, the main isoflavone in soy, has a two-phase effect on the ER present on the breast cancer cells (Figure 4). At low concentrations, it stimulates the growth of positive-ER breast cancer cells, whereas at higher concentrations, it inhibits the growth of breast cancer cells. Another mechanism that may be involved in cancer prevention includes the inhibition of a tyrosine kinase system that can prevent abundant cell proliferation or abnormal angiogenesis by inhibiting signaling pathways associated with tyrosine kinase receptors. In addition, genistein inhibits the activity of DNA topoisomerase [50,51,52].

Various reports on the relationship between soy and its beneficial effects on breast tumors in postmenopausal women are available. A prospective cohort study of 15,607 women carried out in Japan found a lower risk of breast cancer in a group of subjects consuming soy and soy products. After taking into account the pre-and postmenopausal periods, a statistically significant relationship was only recorded in women who were in a postmenopausal period [53]. 

According to a meta-analysis by Chen et al. [24], a negative correlation was found between the consumption of soy isoflavones and the risk of breast cancer in pre-and postmenopausal Asian women. This relationship was not observed in women living in Western countries. One explanation for this discrepancy may be the high intake of isoflavones by Asians not only in adulthood, but also from childhood. In addition, several control studies have reported that the consumption of soy and isoflavones from early childhood may reduce the risk of developing breast cancer later in life [54,55,56]. 

A randomized, double-blind, placebo-controlled clinical trial conducted by Delmanto et al. [57] determined the effect of soy isoflavones on breast density and breast parenchyma assessed by mammography in 80 postmenopausal women. After 10 months of isoflavone supplementation, it was found that the daily intake of 100 mg isoflavones did not affect the breast density (assessed by mammography) and breast tissues (evaluated by ultrasound).

Khan et al. [58] evaluated the effect of soy isoflavone supplementation on breast epithelial proliferation and other biomarkers in healthy high-risk Western women. The breast cell proliferation marker is an intermediary marker of breast cancer risk generally thought to be more reflective of risk than mammographic density. Cells were examined for the Ki-67 labeling index and atypia. The study involved 98 women of a pre- and postmenopausal age. After 6 months of isoflavone supplementation, the authors found that the daily intake of 235 mg of isoflavones did not increase cell proliferation in pre-and postmenopausal women compared to the placebo. In contrast, combined hormone therapy increased breast cell proliferation in postmenopausal women [59].

In their study, Johnson et al. [60] screened 54 soybean cultivars and suggested that the pro- and antiestrogenic activity of the soybean is determined by the isoflavone composition, especially the relative concentrations of daidzin and genistin. They showed that daidzin caused promotion, invasion, and proliferation in breast cancer cells and antagonized tamoxifen, whereas the aglycone genistin opposed it. They also suggested that an in vivo comparison of isoflavones extracts from soybeans grown under identical climate and agronomic practices are required before recommending them as chemopreventive agents for breast cancer and as an HRT alternative for postmenopausal women.

Prostate cancer is the second most commonly diagnosed cancer and the sixth most common cause of cancer death in men worldwide [61]. The incidence of PCa significantly differs between ethnic populations and countries. Incidence rates are the lowest in Asian countries, where soy foods are regularly consumed as part of a normal diet [62,63]. A meta-analysis carried out by Applegate et al. [61] provided an updated, systematic analysis of the available literature describing the relationship between soy food intake and PCa risk. The authors showed a statistically significant relationship between soy consumption and decreased PCa risk. In a population-based case–control study on the association between PCa and the dietary phytoestrogen consumption of southern Italians, the authors found that isoflavones, specifically genistein, were correlated with a reduced risk of PCa [64]. 

Wu et al. [65] evaluated the relationship between genistein in plasma, epidemiological factors, and PCa in the Chinese population. Among the 100 patients evaluated, 46 (46.0%) were diagnosed with PCa. The median plasma genistein concentration of non-PCa patients (728.6 ng/mL) was significantly higher than that of PCa patients (513.0 ng/mL). A high blood concentration of genistein was associated with a 70% reduction in PCa. Based on these results, the authors suggested that the high concentration of plasma genistein may contribute to the low incidence of PCa in the Chinese population. Nagata et al. [66] also suggested that high serum levels of genistein, daidzein, and glycitine are significantly associated with a reduced risk of PCa in the Japanese population. 

## 7. Effect on Cardiovascular Disease

Cardiovascular disease (CVD) is one of the main causes of death in most Western and developing countries. CVD includes all diseases affecting the heart and blood vessels and includes coronary heart disease (CHD), dyslipidemia, hypertension, and coronary artery disease [67]. Soy-based foods have attracted scientific attention since 1999, when the United States Food and Drug Administration (FDA) approved the health claim that 25 g of soy protein daily, along with a diet low in saturated fat, may reduce the risk of CVD. Afterwards, claims on soy were released in other countries such as Canada, the UK, Brazil, Indonesia, and the Philippines, mostly for 25 g of soy proteins as intervention for cardiovascular protection [22,68,69]. Moreover, epidemiological studies indicate that the long-term consumption of soy and soy products is associated with a lower incidence of cardiovascular disease [70]. This hypothesis is also supported by the low rates of cardiovascular diseases in Asian populations, where the diet is particularly rich in soy, followed by the loss of this protection among the groups that have moved to Western societies [71]. A beneficial effect on the lipid profile of food products containing soybeans has been observed, whereas the extracts of isoflavones do not show this effect. It seems that the hypocholesterolemic effect is exerted by soy protein, and isoflavones may favorably affect the endothelium of blood vessels [72].

A meta-analysis of 38 research studies showed a 9.3% reduction in total cholesterol (TC), 12.9% in low density lipoprotein (LDL), and 10.5% in triglycerides (TAG). In these studies, the average consumption of soy protein was 47 g per day. The greatest decreases in cholesterol were seen in those with the highest starting levels [73]. Another meta-analysis carried out by Zhan and Ho [74] showed that the intake of soy protein containing isoflavones significantly reduced serum total cholesterol, LDL-cholesterol, and triglyceride, and significantly increased high density lipoprotein (HDL) cholesterol, but the changes were related to the level and duration of intake and the sex and initial serum lipid concentrations of the subjects. A recent meta-analysis of 35 randomized controlled studies reported by Tokede et al. [69] concluded that isoflavone supplementation had no effect on the serum lipid profile. Moreover, their report showed that an intervention with soy proteins increases the serum HDL concentration and lowers serum TAG, LDL, and TC concentrations. A study carried out by Sathyapalan el al. [6] has shown that six-month administration of 15g soy protein per day with or without 66 mg isoflavones to early menopause woman did not change the total cholesterol, LDL, HDL, or triglyceride levels.

It seems reasonable to conclude that current evidence from randomized control trials in humans does not sufficiently support the claim that isoflavone extracts can independently reduce CVD risk by modulating plasma lipids. It is possible that soy isoflavones may reduce CVD risk by protecting against the oxidation of LDL cholesterol and the development of oxidized LDL, as opposed to having a lipoprotein-lowering effect [70].

Several studies suggest that the effect of isoflavones on endothelial function may be related to an individual’s capacity to metabolize daidzein into equol [22,70]. Recently, a double-blind crossover study demonstrated significant improvements in arterial stiffness, blood pressure, and endothelial function with the consumption of purified equol supplements, but only among equol-producing men [75]. The blood pressure-lowering effect of isoflavones may also be attributed to the activation of endothelial nitric oxide synthase (eNOS) and the stimulation of nitric oxide (NO) production [76]. A study has suggested that isoflavones attenuate blood pressure elevation through the acceleration of NO production and inhibition of inflammation [77].

A meta-analysis of 11 studies concluded that soy isoflavones had the effect of lowering the blood pressure in hypertensive subjects, but not in normotensive subjects [78]. Another meta-analysis as reported by Beavers et al. [79], indicated that exposure to soy isoflavones can modestly, but significantly, improve endothelial function, as measured by flow-mediated dilation. Malek Rivan et al. [68] investigated the association between the intake of soy isoflavones and blood pressure among multiethnic Malaysian adults. The author suggested that soy protein and soy isoflavones intake had a significant association with blood pressures. Soy protein intake showed a significant association with diastolic blood pressure, while soy isoflavone intake showed a significant association with systolic blood pressure (SBP). It is known that blood pressure, especially SBP, is important as an independent risk factor for coronary events, stroke, heart failure, and chronic kidney disease [68]. Additional epidemiological studies are needed to strengthen evidence on the protective effect of soy on cardiovascular disease risk factors.

## 8. Conclusions

Soy and its products seem to be an ally in the prevention of civilization diseases. However, the results of epidemiological studies on the relationship between soy consumption and the prevention of diseases are still incomplete. The largest group of people who have been studied in this respect are the residents of Asian countries, where soybeans, next to rice, represent the major component of their daily meals. The results of these studies cannot be easily applied to European societies because there are too many differences in dietary, lifestyle, environmental, and genetic factors. Therefore, studies on the health-promoting properties of soy products are still being developed. However, it is undeniable that soy products should be included in the daily diet.

## Figures and Tables

**Figure 1 nutrients-11-01660-f001:**
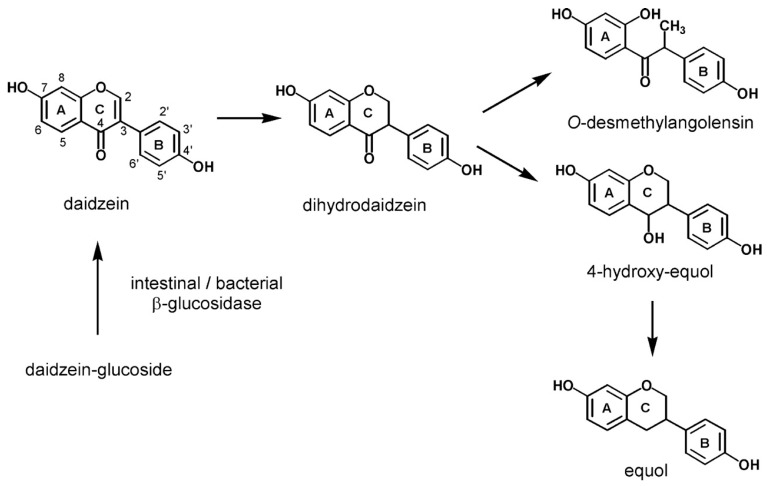
Formation and metabolism of daidzein [15].

**Figure 2 nutrients-11-01660-f002:**
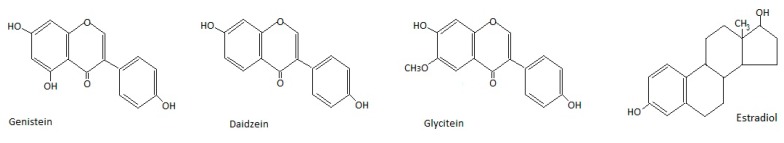
Chemical structures of isoflavones and 17-β-estradiol.

**Figure 3 nutrients-11-01660-f003:**
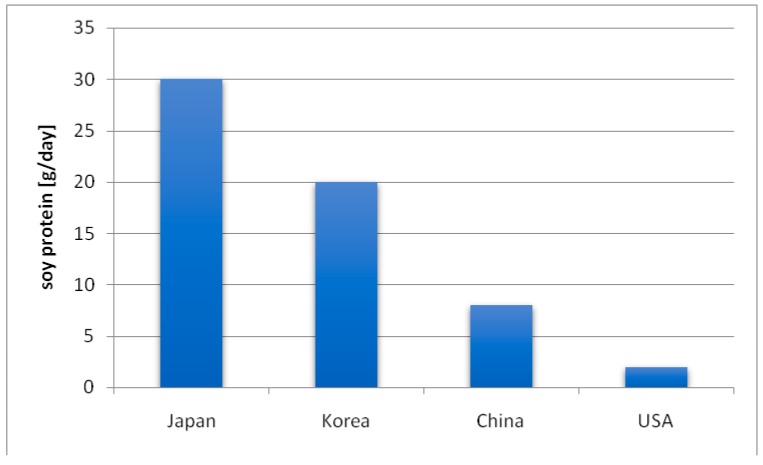
Soy consumption across different populations [20].

**Figure 4 nutrients-11-01660-f004:**
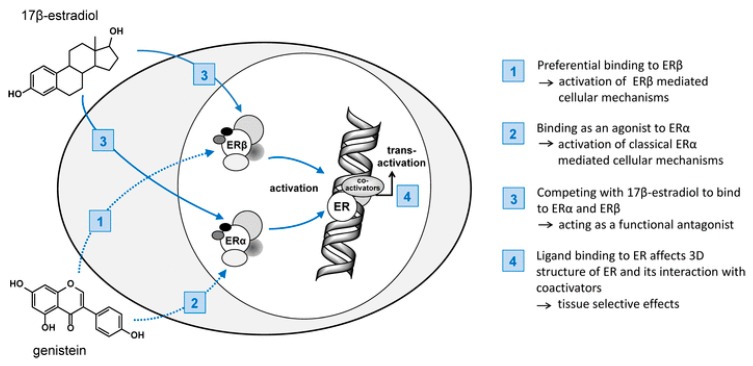
Mechanism of action of 17β-estradiol and potential interaction of isoflavones in the cascade. ER, estrogen receptor; SERM, selective estrogen receptor modulator [11].

**Table 1 nutrients-11-01660-t001:** Observational studies on the effect of isoflavone intake and risk of diabetes.

Model	Number of Participants	Duration	Isoflavone Intake	Result	Reference
Vietnamese adults aged 40–65 years	Case: 599 people with newly diagnosed T2DM Control: 599 people in the hospital	24 months	5.2–9.8 mg daidzein/day	Reducing risk of T2DM	[30]
Chinese women	Case: 80 T2DM women Control: 40 women	2 months	435 mg/day	Exposure ofT2DM women to isoflavone supplementation showed reduced risk of diabetes	[31]
Postmenopausal Chinese women, aged 48–62 years.	Mild-dose *n* = 68, High-dose *n* = 67, Control *n* = 68	12 months	Mild-dose 40 mg/day; High-dose 80 mg/day (daidzein 46.7%)	Beneficial effect on reducing fasting glucose by higher doses of daidzein supplementation	[32]
American obese women and men with diagnosed and pharmacologically treated T2DM	104 male and female	12 months	Individual nutrition plan versus 3-1 soy-based meals during the day	plasma glucose concentration reduction by 26.17 mg/dL at 6 months but not at 12 months, HbA1c reduction by 0.49 ± 0.22% at 3 and 6 months, but not at 12 months	[33]

T2DM—type 2 diabetes; HbA1c—glycolized hemoglobin.

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
