# Peer review of "Biological Effect of Soy Isoflavones in the Prevention of Civilization Diseases"

_nutrients, 2019, doi:10.3390/nu11071660_

Reviewer 1 Report

The manucript has been improved in the revised version.

Reviewer 2 Report

I am satisfied with the revisions. Thanks for your efforts.

This manuscript is a resubmission of an earlier submission. The following is a list of the peer review reports and author responses from that submission.

Round  1

Reviewer 1 Report

The proposed article is well organized and presented in a nice way.

Here are some corrections/proposals for authors.

1)    Line 28: “conditions” Please rewrite

2)    Line 53: Please delete “polyphenolic compounds that are”

3)    Figure 3: Please put scales, numbers and names

4)    Line 111: Please delete the second “receptors”

5)    Line 113: Please rewrite “and the latter two processes” there is a misunderstanding

6)    Line 141-142: Please rewrite the sentence – “at urine sample collection”

7)    Line 174: “)” please delete

8)    Please add some new studies 2018-2019 with the analogous references.

9)    Please check carefully the references list in order to be according to journal style, for example some journals’ names are not abbreviated

Reviewer 2 Report

 Reviewer's Comments:

·        Page 1, Line 10

o   Would be helpful to have these diseases defined as examples of civilization disease after the 1st sentence in abstract instead of waiting towards the end.

·        Page 2, Line 47

o   You mean "against diabetic condition" not "against antidiabetic conditions" which is double negative.

·        Page 2, Line 51

o   Other useful information you can optionally provide in this paragraph to enrich our understanding may include the fact that "Isoflavones" are sometimes referred to as "Phytoestrogen" but the former is preferred over the latter term.

·        Page 3 Lines 85 and 86

o   The X axis labels (US, Western Countries, South East Asia) are missing in Figure 3. A fuller descriptive Figure Footnote would be very helpful.

·        Page 7, Line 259

o   A section dedicated to Isoflavones and CVD, Hypertension is warranted because there are many references on this connection. It is currently missing.

*** See attached edits

Reviewer 3 Report

The manuscript entitled “Health potential of soy isoflavones in the prevention of civilization diseases” by Pabich and Materska reviewed health effects of soyisoflavones in life style diseases. While this is a well written review, the following major concerns needs to be addresses.

Major Concerns:

1. While text of the manuscript has discussed both beneficial and deleterious effects of soy isoflavones, title and abstract of the manuscript giving the impression that soy isoflavones has only health beneficial effects. This is misleading, therefore, suggest to modify the Title and Abstract to reflect the right balance of health beneficial and deleterious effects of soy isoflavones.

2. Some critical studies and findings are omitted from the review and including them will strengthen the manuscript. For example, it has been shown that composition of isoflavones is as important as isoflavone content (Kuiper et al., Interaction of estrogenic chemicals and phytoestrogens with estrogen receptor beta. Endocrinology. 139, 4252–4263 (1998) and Johnson et. al., Glycone-rich Soy Isoflavone-Extracts Promote Estrogen Receptor Positive Breast Cancer Cell Growth. Nutrition and Cancer, 2015.) Johnson et. al., has screened soybean cultivars and shown the paradoxical effects of daidzin, a glycone in promoting breast cancer cell growth and antagonizing tamoxifen where as the aglycone genistin in opposing it.

3. Figure 3 is missing figure legends as well as labels for bars; without these it is difficult to interpret.

4. Table 1, should provide numerical values to reflect an increase or decrease of Hb1AC and/glucose levels etc.

5. Figure 4, distinguish the antagonistic and agonistic pathways with different shade or color.

Reviewer 4 Report

The authors summarize current knowledge on the potential health effects of soy isoflavone consumption in the civilization diseases (e.g. osteoporosis, type II diabetes, hypercholesterolemia, cancer). The paper in easy to be read and well written. Fig 3 must be better presented in order to be self-explanatory. The authors could consider to add other tables to illustrate referenced studied they evaluated (cross-sectional or longitudinal studies, animal studies and additionally in vitro studies) according to single disease.